# Influence of Oxidative Stress and Inflammation on Nutritional Status and Neural Plasticity: New Perspectives on Post-Stroke Neurorehabilitative Outcome

**DOI:** 10.3390/nu15010108

**Published:** 2022-12-26

**Authors:** Irene Ciancarelli, Giovanni Morone, Marco Iosa, Antonio Cerasa, Rocco Salvatore Calabrò, Giovanni Iolascon, Francesca Gimigliano, Paolo Tonin, Maria Giuliana Tozzi Ciancarelli

**Affiliations:** 1Department of Life, Health and Environmental Sciences, University of L’Aquila, 67100 L’Aquila, Italy; 2ASL 1 Abruzzo (Avezzano-Sulmona-L’Aquila), 67100 L’Aquila, Italy; 3San Raffaele Institute of Sulmona, 67039 Sulmona, Italy; 4Department of Psychology, Sapienza University of Rome, 00185 Rome, Italy; 5IRCCS Fondazione Santa Lucia, 00179 Rome, Italy; 6Institute for Biomedical Research and Innovation (IRIB), National Research Council of Italy, 98164 Messina, Italy; 7Pharmacotechnology Documentation and Transfer Unit, Preclinical and Translational Pharmacology, Department of Pharmacy, Health Science and Nutrition, University of Calabria, 87036 Calabria, Italy; 8S. Anna Institute, 88900 Crotone, Italy; 9IRCCS Centro Neurolesi “Bonino-Pulejo”, 98123 Messina, Italy; 10Department of Medical and Surgical Specialties and Dentistry, University of Campania ‘Luigi Vanvitelli’, 80138 Naples, Italy; 11Department of Mental and Physical Health and Preventive Medicine, University of Campania ‘Luigi Vanvitelli’, 80138 Naples, Italy

**Keywords:** nutritional status, malnutrition, stroke, oxidative stress, rehabilitation, neuroplasticity, healthy dietary patterns

## Abstract

Beyond brain deficits caused by strokes, the effectiveness of neurorehabilitation is strongly influenced by the baseline clinical features of stroke patients, including a patient’s current nutritional status. Malnutrition, either as a pre-stroke existing condition or occurring because of ischemic injury, predisposes patients to poor rehabilitation outcomes. On the other hand, a proper nutritional status compliant with the specific needs required by the process of brain recovery plays a key role in post-stroke rehabilitative outcome favoring neuroplasticity mechanisms. Oxidative stress and inflammation play a role in stroke-associated malnutrition, as well as in the cascade of ischemic events in the brain area, where ischemic damage leads to neuronal death and brain infarction, and, via cell-to-cell signaling, the alteration of neuroplasticity processes underlying functional recovery induced by multidisciplinary rehabilitative treatment. Nutrition strategies based on food components with oxidative and anti-inflammatory properties may help to reverse or stop malnutrition and may be a prerequisite for supporting the ability of neuronal plasticity to result in satisfactory rehabilitative outcome in stroke patients. To expand nutritional recommendations for functional rehabilitation recovery, studies considering the evolution of nutritional status changes in post-stroke patients over time are required. The assessment of nutritional status must be included as a routine tool in rehabilitation settings for the integrated care of stroke-patients.

## 1. Introduction

Relevant reports highlight that despite the implementation of effective prevention strategies aimed at reducing the influence of risk factors and the consequences of a high economic burden for society due to stroke-related long-term sequelae, which are often the cause of severe disability, the incidence of strokes, namely, ischemic strokes, will realistically increase worldwide as a new or recurrent event [1,2,3,4,5]. Indeed, hemiparesis, paretic muscle atrophy, and spasticity, together with frequent cognitive impairments, lead to the development of permanent neurological disability resulting in unsatisfactory and limited independence in activities of daily living. The brain’s self-repair mechanisms and most of the mechanisms of neuronal plasticity, which enable the reorganization of damaged neuronal circuits, even among elderly people, play a key role in the recovery of neurological deficits. Neurorehabilitation is still the gold-standard stroke treatment for reducing functional disability by promoting new functional communications in the remaining neuronal circuits and enhancing neuronal activity in damaged networks [6,7,8]. Beyond the brain deficits brought on by the ischemic insult, the pre-existing baseline clinical features of stroke patients—including a patient’s current nutritional status, constituting an undeniable index of their general health status—have a significant impact on the effectiveness of a rehabilitation treatment [9]. Old age, cognitive impairment, depression, insufficient physical activity, and comorbidities represent risk factors favoring a poor nutritional status/malnutrition in stroke patients [10,11]. In addition, in patients who are not malnourished prior to suffering a brain ischemic event post-stroke, dysphagia, hemiparesis, reduced mobility, and psychiatric diseases can exacerbate poor nutritional status/malnutrition existing prior to the stroke’s occurrence, thereby severely worsening functional recovery by establishing malnutrition [12,13,14]. On the other hand, a proper nutritional status, compliant with the specific and urgent needs required by the process of brain recovery, plays a key role in post-stroke rehabilitative outcome favoring neuroplasticity mechanisms [14,15,16,17,18,19,20,21,22]. Therefore, the assessment of nutritional status is of remarkable interest in stroke patient care, and the programming of specific nutritional interventions essential to controlling risk factors, as well as scanning for the actual presence of malnutrition due to its negative effects on rehabilitative outcomes, should be prioritized. However, despite the widespread agreement that stroke patients’ nutritional status is a crucial factor to achieving a satisfactory clinical recovery both during and after rehabilitation, the studies that are currently available do not allow to draw conclusive results about the mechanisms involved in the patient’s nutritional status examined before and after rehabilitation treatment. Additionally, nutritional status assessment scales are not typically included in the clinical scales used to evaluate rehabilitation outcomes. Hence, the sharp comment made by Engelhardt et al. [23] that nutrition is still inadequately integrated into health systems is very relevant. To elaborate nutritional recommendations regarding post-stroke care and functional rehabilitation recovery, studies that consider the evolution of the changes in the nutritional status of stroke patients over time are required.

The current narrative review focuses on the available evidence regarding the involvement of redox balance and inflammation in the nutritional status of post-stroke patients and their influence on neuroplasticity processes and rehabilitative outcomes. 

## 2. Nutritional Status and Healthy Brain

The World Health Organization (WHO) defined nutritional status “as the condition of the body, resulting from the balance of intake, absorption, and utilization of nutrients interacting with individual physiological and pathological status” [24,25,26]. Accordingly, nutritional status, which is measurable at a given time, in different age ranges, and in physiological or pathological conditions, represents the result of the interaction of the intake, absorption, and utilization of macro- and micronutrients (Figure 1).

Bodily functions measured through biochemical parameters provide useful insights, primarily into the utilization of macro- and micronutrients according to a specific function. The energetic balance and the balance of any single nutrient can be used to estimate short-term changes in nutritional status and the body composition represents its long-term index. Therefore, the tools required for an adequate nutritional assessment should include eating habits/dietetic reports, anthropometric and functional indices, and the assessment of nutritional risk factors. However, due to the many variables involved in assessing nutritional status, no single set of accepted standards for its assessment is available. As a result, to identify risk factors for malnutrition that are known to be associated with poor rehabilitative recovery, functional impairment, a reduced quality of life, and increased morbidity and mortality, the assessment of the nutritional status of stroke patients remains complex but necessary in clinical practice [27,28]. Among the most-used tools [29,30,31], the Mini Nutritional Assessment (MNA) and the MNA short form (MNA-SF) are valid in identifying nutritional risk and may be useful to clinicians for developing interventions to improve the nutritional status of patients. Recent studies [32,33,34,35] support the use of the Controlling Nutritional Status (CONUT) score—developed by Ignacio et al. [31]—and the Prognostic Nutritional Index (PNI) score as sensitive screening tools for nutritional status assessment. Both these rating scales could help to identify patients who would benefit from early nutritional therapy. There is a general agreement that dietary patterns, rather than a single bioactive substance, nutrient, or their groups, can impact the nutritional status of any individual, regardless of gender, age, and race, and exert beneficial or negative effects, which are either immediate or long-term [36,37,38]. As any other organ, the brain utilizes nutrients from foods, and evidence shows that several nutritional factors related to one’s eating habit can affect the structure and function of the brain [39]. Appropriate lifestyle behaviors, including maintaining proper nutrition and regular exercise, are essential first steps in avoiding many diseases, including ischemic strokes, and their harmful effects. Findings have confirmed that unhealthy dietary habits increase the burden of the risk factors in the onset of many diseases by increasing the susceptibility to oxidative stress and inflammation, which are strongly interrelated [40,41,42,43]. 

### Oxidative Stress, Inflammation, and Healthy Nutrition

Oxidative stress occurs because of an imbalance between the production of free radicals (reactive oxygen species—ROS—and reactive nitrogen species—RNS) and the antioxidant system’s effectiveness/availability. Free radicals are short-living reactive chemical species that, under physiological conditions, remain confined to their site of formation because of their short life span and endogenous antioxidant defense mechanisms that counteract their progression and damaging oxidative effects. A disturbance in this balance leads to oxidative stress, which causes damage to important macromolecules such as DNA, proteins, and lipids, resulting in the production of oxidative metabolites that persist in the systemic circulation as critical elements with multiple functional targets [43,44,45,46]. As is widely known, a strong relationship exists between oxidative stress and inflammation. In a sort of reverberating and vicious circle, the excessive availability of circulating oxidative metabolites can activate inflammatory-signaling pathways, while inflammation induces oxidative stress [43,45] (Figure 2). 

Healthy eating habits can maintain and control balanced crosstalk between oxidative stress and inflammation, thereby assuming a fundamental role in clinical care. 

Proper dietary patterns based primarily, but not exclusively, on vegetables—such as the Mediterranean (MD) or Nordic diets, the Dietary Approaches to Stop Hypertension Trial (DASH) diet, and the Mediterranean-DASH Intervention for Neurodegenerative Delay (MIND) diet—may support the achievement of a proper nutritional status [46,47,48,49]. The apparent health benefits of the foods at the center of these eating habits (legumes, fruits, vegetables, fish, and extra virgin olive oil) and of a general healthy diet can be ascribed to the properties of a variety of active compounds with complex chemical structures (phytochemicals, vitamins, and poly-unsaturated fatty acids, etc.). Active compounds work as powerful synergic modulators of oxidative stress and inflammation [50]. Beyond heart disease, effective nutritional strategies for health, at both primary and secondary levels of prevention, and disease management, such as that for strokes, cognitive impairment, depression, and neurodegenerative diseases, are—or should be—significantly based on stemming inflammation and restoring redox balance by managing circulating levels of inflammatory and oxidative stress metabolites and boosting antioxidant defenses [36,40,42,43]. Studies have emphasized the promising influence of healthy dietary habits and a resultant proper nutritional status on neuroplasticity, which represents the ability of a healthy brain to undergo structural and functional changes in response to internal bodily and environmental changes and damage. Accordingly, findings underline the positive effects of the omega-3 (n-3) fatty acids docosahexaenoic acid (DHA) and eicosapentaenoic acid (EPA), contained mainly in fish and nut oils and in leafy vegetables, towards improving neurotransmission and modulating cholesterol-induced decreased membrane fluidity, which are fundamental to supporting cell signaling and synaptic plasticity [51,52,53]. Evidence shows that the regular consumption of polyphenol-rich foods can promote neurogenesis, neurodevelopment, and synaptogenesis, as well as have positive effects on cognition and cerebral blood flow [54,55,56]. Recently, a potential influence on neuroplasticity, neurogenesis, and cerebral blood flow was attributed to flavonoids, a complex family of compounds present in many edible plants with well-known antioxidant and anti-inflammatory activity [57,58,59,60]. Interestingly, attention was focused on the effects of trans-resveratrol, a polyphenolic phytoalexin present in many consumed foods, on neuroplasticity, neurogenesis, and neuroprotective mechanisms. 

In addition to an increased expression and activity of endogenous antioxidants such as catalase, superoxide dismutase, and glutathione peroxidase, such significant effects are mediated by the activation of nuclear erythroid 2-related factor 2, a regulator of cellular resistance to oxidants that controls the basal and induced expression of an array of genes dependent on the response to oxidative injury [61].

Therefore, it is clear that dietary elements influencing the crosstalk between inflammation and oxidative stress (e.g., MD) may provide neuroprotective effects and promote the activation of pathways involved in the mechanisms underlying the molecular mechanisms of neuronal plasticity. Overall, dietary patterns rich in health-promoting foods, combined with other aspects of daily living, such as physical activity, positively impact nutritional status and health outcomes, thus guaranteeing the optimal conditions for maintaining a healthy brain.

## 3. Ischemic Stroke and Nutritional Status

There is a consensus on the importance of evaluating the nutritional statuses of patients who have suffered an ischemic stroke in order to have a full picture of their clinical profile, to check for early malnutrition or its pre-existence, and to identify the main factors involved in worsening nutritional status, with the main goal of developing a rehabilitative strategy tailored to patients’ neurological deficits and medical history. Indeed, malnutrition resulting from a deficiency or from an excess of nutrients predisposes patients to poorer functional outcomes following a stroke. In fact, the fundamental role of the common and final denominators of factors, both interconnected with one another and not, related to pre-existing stroke comorbidities and acute ischemic sequelae explains the lack of recovery [9,10]. Malnutrition, which is frequently observed in stroke patients, is of a multifactorial origin, and may depend on the patient’s pre-existing stroke diseases, as assessed by their medical history; on consequences directly resulting from a stroke-induced systemic cascade of events that is synergistically responsible for metabolic adaptive and maladaptive changes; and on both conditions [62]. 

A patient’s age should also be considered, as the elderly are more susceptible to strokes and undernutrition/malnutrition. Indeed, it is well known that the mechanisms orchestrating nutrient intake (hunger, satiety, and thirst) tend to change with advancing age, so changes in body composition (body mass index (BMI), anthropometric measures, fat mass, and fat-free mass) and in several functions (hormones, gastrointestinal motility, the immune system, etc.) are further altered in old stroke patients [12,14]. Dysphagia, along with all its associated symptoms and consequences, represents the most important cause of a reduced intake of nutrients and of increased susceptibility to changes in the nutritional status of stroke patients [14]. Furthermore, neurological deficits, mild or severe cognitive impairment, and stroke-induced catabolic processes contribute to promoting the difficulty of oral feeding and, above all, rendering it insufficient. Pre-existing stroke conditions such as unhealthy lifestyle behaviors, poor socio-economic conditions, the presence of chronic diseases, and polypharmacy also lead to malnutrition, namely, to protein-energy malnutrition [63]. Notably, an ischemic brain injury induces or exacerbates diseases, which, in part, may depend on insufficient/incorrect nutrition, such as anemia, altered plasma glucose levels, hypocalcemia, and sarcopenia, for which evaluation is important with respect to their negative influences during and after rehabilitation [21,22,64]. 

### 3.1. Malnutrition and Stroke-Modified Cross-Talking between Oxidative Stress and Inflammation

Free radicals such as superoxide, nitric oxide, and hydroxyl radicals, and other reactive species such as hydrogen peroxide, peroxynitrite, and hypochlorous acid, when over-produced, as in the case of an ischemic stroke, start complex chain reactions leading to the production of oxidative metabolites capable of maintaining oxidative stress in the systemic circulation and thus interfering as critical factors with multiple functional targets [43,44]. A wide array of enzymatic and non-enzymatic compounds acting as antioxidants have been ascribed to play a crucial role by synergically cooperating to maintain the homeostasis of redox balance and inflammation [44]. The neuronal redox state represents a particular model of homeostatic equilibrium that is established between the generation of radicals and the antioxidant defense. Brain cells are susceptible to the harmful effects of oxidative injury because neuronal membrane components are rich in polyunsaturated fatty acids, highly susceptible to oxidation, require large amounts of oxygen for energy production, and are relatively poor with respect to antioxidant defenses, which are mostly localized in glia cells [65]. The ischemic injury triggers a fast excessive increase in the generation and release of free radicals, such as superoxide anions (ROS) and reactive nitrogen species (RNS), and a simultaneous depletion of endogenous antioxidant defenses, thus activating cellular pathways responsible for neuronal necrosis and apoptosis [65,66,67,68]. Indeed, it is known that superoxide dismutase (SOD) is responsible for scavenging superoxide radicals, and that glutathione peroxidase (GPx), a selenocysteine-dependent enzyme, is the most important hydrogen peroxide-scavenging enzyme. SOD and GPx can directly counterbalance an oxidative attack and protect the cells against DNA damage [43,46,64]. A noteworthy point is that uric acid functions as an important endogenous antioxidant in blood plasma where it represents more than half of the plasma total antioxidant capacity and may play a beneficial role in endothelial function by preventing the degradation of Cu/Zn superoxide dismutase, an extracellular enzyme that belongs to the family of oxidoreductases and prevents nitric oxide (NO) decomposition by scavenging superoxide anions [69]. Research findings have demonstrated the importance of the intake of minerals to decreasing inflammatory and oxidative stress, and thereby their involvement in decreasing the risk for several diseases. Consequently, a nutritional deficiency of minerals such as copper, calcium, zinc, iron, selenium, and magnesium may cause the inappropriate function of endogenous antioxidant enzymes (e.g., selenium deficit and GPx activity; copper and zinc deficits; and Cu/Zn superoxide dismutase), thus contributing to the long-lasting cytotoxic effects of free radicals and to the development of an inflammatory status. According to a recent study, disturbed mineral homeostasis may play a significant role in the pathogenesis of ischemic stroke, and the significant disruption of the serum Cu/Zn and Cu/Se molar ratios could be as a sensitive indicator of oxidative stress and nutritional status of stroke patients [70]. Magnesium deficiency worsens inflammation and oxidative stress and raises the risk of several diseases, including strokes, especially in the elderly [71]. Zinc has been identified to play a significant part in the regulation of nutrition via its involvement in appetite mechanisms. In fact, experimental evidence showed that dietary zinc deficiencies impair appetite and food intake and decrease body mass [72]. Therefore, it may be possible to modulate the molecular mechanisms regulating inflammation and redox balance by intervening to maintain proper mineral availability through dietary supplementations and/or pharmacological interventions. There is evidence that oxidative stress elicited during the initial phase of cerebral ischemia initiates the signaling of molecular sequelae leading to the activation of transcription factors and proinflammatory gene expression that secrete inflammatory cytokines [66,69,71]. The development of brain-resident cell-mediated inflammation is the main source of oxidative stress after a stroke’s onset and promotes the expansion of the ischemic lesion, blood–brain barrier dysfunction, and systemic inflammatory response syndrome, which negatively affect the stroke’s outcome [63,68,73,74]. 

### 3.2. Stroke-Induced Modifications of Gut-Brain Axis: Involvement of Oxidative Stress and Inflammation

The systemic inflammatory response occurring after an ischemic insult is a complex and articulated mechanism in which inflammation and oxidative stress concur to induce dysfunction in other organs and systems. The gut microbiome influences intestinal barrier function and facilitates the maintenance of the immune, nutritional, and metabolic homeostasis that influences appetite and food intake, thereby preserving a proper nutritional status [75,76,77]. Several reports have shown the impact of ischemic stroke on the gut microbiome and the consequence of intestinal dysbiosis mediated by oxidative stress and inflammation [78,79,80]. Stroke-induced alterations in the gut–brain axis compromise gut microbiome metabolites that influence intestinal barrier, appetite, and satiety-regulating systems, thus precipitating or worsening malnutrition. Consuming nutrients with properties that counteract free radicals such as fruit, vegetables, and legumes, along with maintaining a proper fiber intake, may enhance the gut microbiome and intestinal peristalsis, which concur to normalize the acid–base balance and reduce the production of proinflammatory cytokines [80]. From the results, the gut microbiome could be a therapeutic target for an effective management of nutrition of stroke patients and a valid support for the treatment of malnutrition. The available findings have long proven the relationship between several cytokines and malnutrition [81]. Recently, it has been reported that serum levels of interleukin-18 (IL-18), which is involved in the mechanisms controlling food intake by appetite regulation, acutely ill older hospitalized. This suggest that interleukin-8 (IL-8) may play a role in malnourished stroke patients—who are often elderly, too [82]. Consistent with the association between inflammation and malnutrition, evidence has been provided concerning an inverse correlation between BMI in malnourished, healthy individuals and increased plasma levels of tumor necrosis factor α (TNF-α) [83]. It is interesting to note that severely malnourished patients’ neutrophils have a reduced capacity to fight infectious illness by producing ROS [83]. Moreover, it is conceivable that an impaired ability to preserve immune homeostasis by balancing the production of oxidative and pro-inflammatory mediators may contribute to the increased susceptibility of malnourished stroke patients to infections, constituting the most common complication after stroke, which worsens clinical recovery. 

Therefore, the modulation of the cross-talk between oxidative stress and inflammation induced by brain ischemic injury could be a strong and modifiable factor involved in the malnutrition of patients after a stroke and, consequently, in the global prognosis of stroke patients (Figure 3).

## 4. Neural Plasticity and Stroke-Induced Redox Imbalance and Inflammation

The brain’s undoubted need for a high quantity and uninterrupted availability of energy substrates, together with the lack of an energy storage system, can account for the functional significance of the neurovascular unit (NVU), a complex functional structure for communication between brain cells and cerebral vessels whose function has long been known and was confirmed as “the symbiotic relationship between brain cells and cerebral blood vessels, calling attention to their developmental, structural and functional interdependence in health and disease” at the 2001 Stroke Progress Review Group meeting of the National Institute of Neurological Disorders and Stroke [84]. The blood–brain barrier (BBB) and neuronal homeostasis are maintained in physiological conditions through the proper crosstalk between the NVU’s components parts. Glial cells are heterogeneous and highly specialized cells integrated in many mechanisms for preserving brain homeostasis, for which examples include the role of astroglia in providing nutritional substances and microglia/macrophages’ ability to carry out immune responses against brain damage [85,86,87]. As in other cells, the normal respiratory metabolism of brain cells results in the production of ROS, which comprise free radical and non-radical/molecular forms. This is particularly true of the mitochondria but it also occurs in other subcellular structures such as the plasma membrane, peroxisome, cytosol, endoplasmic reticulum, and extracellular space [88,89,90,91,92]. Moreover, neuronal and endothelial nitric oxide synthases (nNOS and eNOS, respectively) generate RNS in the form of NO, a gaseous radical involved in vascular homeostasis, neuronal signaling, and brain plasticity, and highly reactive peroxynitrite, which is produced by the interaction of nitric oxide with O_2_^−^ [93,94]. Although the brain is relatively poor of antioxidant defense, which is a complex system of constitutive and non-constitutive antioxidant defenses that cooperate to maintain brain redox homeostasis [95], enzymes such as superoxide dismutases, catalases, thioredoxin reductases, glutathione peroxidases, and non-enzymatic molecules (ROS scavengers) available in the cell via synthesis or diet (e.g., alpha-tocopherol, ascorbic acid, β-carotene), mediate a static antioxidant defense. Pathways such as Nrf2-ARE (nuclear factor erythroid 2-related factor (Nrf2), which is a transcription factor recognizing the antioxidant response element (ARE), predominant in glial cells) and the JNK/AP-1 system (the c-Jun-N-terminal Kinase (JNK) pathway is a ‘death’-signaling pathway controlling cellular responses to harmful extracellular stimuli; AP-1 is a transcription factor controlling several cellular processes including differentiation, proliferation, and apoptosis) are involved in the adaptive antioxidant defense and promote the transcription of genes encoding antioxidant response proteins [95,96]. 

### Neuronal Redox Status: A Dynamic Mechanism for Maintaining Adequate Neuronal Cell–Cell Signaling in Healthy Brains and after Ischemic Stroke

The hypothesis that ROS and RNS represent keystone mechanisms regulating a variety of physiological processes ranging from neuronal development and structural adjustments to synaptic transmission and neuroplasticity, as well as triggering oxidative injury within the nervous system, is of current and particular interest. The involvement of ROS in physiological conditions and the overall redox balance in the regulation of cytoskeletal modification, neural polarity, signaling modulation, and synaptic transmission has been demonstrated [95]. The redox equilibrium is also known to be implicated in the physiological processes of neuroplasticity such as neurogenesis, synaptogenesis, and neurochemical changes of the central nervous system (CNS), which express the healthy brain’s ability to adapt its structures and functions in response to environmental changes, or the need to learn new skills through forming new synaptic connections and deconstructing old ones [96]. Evidence has been provided for the role of ROS/RNS following neuronal activity and in the long-lasting increase in synaptic efficiency, the long-term potentiation (LTP) and long-lasting decrease in the strength of synaptic transmission, and long-term depression (LTD) [97]. Overall, the neuronal redox equilibrium is designed as a dynamic mechanism which is an integral part of the physiological processes preserving homeostasis and the complex functions of nervous system. Current findings reinforce the results of studies carried out over the last decade and emphasize the role of a controlled generation and release of ROS/RNS, which is operated by all the complex subcellular structures of neurons, in the processes of neuronal remodeling, intracellular signaling, synaptic transmission, and the communication between neurons and glial cells, and open the stage for the development of novel, interventional, protective procedures that many natural compounds might potentially promote by interfering with specific sites of the signaling pathways involved in brain plasticity [57,98,99,100]. In a damaged brain, as in ischemic stroke, the critical conditions brought about by neuronal degeneration, edema, and inflammation strongly influence the brain’s capacity to restore the lost functions previously performed by the damaged area, and spontaneous recovery proceeds with compensatory plasticity mechanisms based on the reorganization of neural circuits, new functional communications in the remaining neuronal circuits, and enhancing neuronal activity in pre-existing damaged networks [101]. After an ischemic injury, the NVU loses its function, BBB permeability is enhanced, and multiple mechanisms trigger a cascade of events starting from the decrease in blood flow in the ischemic core until the processes that concur to repair damaged neurons are halted. In the penumbra zone surrounding the ischemic core, the rapid rise in ROS and RNS generation caused by the acute phase of ischemic injury overwhelms antioxidant defenses, resulting in a substantial redox unbalance that is further exacerbated upon reperfusion. The consequent oxidative-nitrosative stress and inflammatory cytokines released by activated microglia and astrocytes critically affect various structural and functional targets, which also predisposes the sufferer to an increased susceptibility of stroke recurrence and other cardiovascular events [101]. Currently, there is general agreement that certain post-ischemic stroke mechanisms may play an apparently opposite role, which is either beneficial or harmful. ROS and RNS, particularly NO, are usually considered neurotoxic chemical species exerting their detrimental effects via the oxidation of essential macromolecules (DNA, RNA, and proteins) and lipid peroxidation [44]. Under a steady-state equilibrium between oxidants and antioxidants, as occurs in healthy conditions, ROS and NO may represent suitable mediators for accomplishing the signaling functions required to maintain adequate cell–cell signaling within the NVU [95,96,97]. In ischemic conditions, as occur in ischemic stroke, the oxidative stress due the overwhelming of the ROS and NO associated with lowered antioxidant potential may provide an adequate environment for the regeneration and repair of the damaged NVU, thus mediating parallel processes of neuroplasticity [102]. The post-ischemic reorganization of cortical representational maps that requires a long-lasting increase in synaptic efficiency (LTP) may explain the apparently controversial role of ROS/RNS, in which the oxidant long-lasting cytotoxic mechanism for both exacerbating neuronal damage (clearance of debris and dead cells) and mediating remodeling cell–cell signaling (neurogenesis and angiogenesis) is essential. It is interesting to note the feedback loop between excessive ROS production and reduced antioxidants availability. ROS generation is somehow allowed by weak antioxidant defense, which is then required for the radicals to burst during the electron transfer step of oxidative metabolism. In this regard, it is intriguing to note that Nrf2 activation is induced by excessive ROS production after a stroke, and that Nrf2 protects the brain against ischemia/reperfusion injury primarily by inducing its target antioxidant genes to counteract excessive ROS production [103]. Experimental findings have shown that the neuroprotective influence of resveratrol is associated with the activation of the Nrf2 pathway by increasing the expression and activity of superoxide dismutases, glutathione peroxidases, catalase, and reducing lipid peroxidation in brain tissue. Furthermore, the activation of the Nrf2 pathway induced by resveratrol administration inhibits neuroinflammation, apoptosis, oxidative stress, and strokes [104,105]. A recent review underlined the possible mechanisms by which peripheral immune components may influence neuronal repair after a stroke, and emphasis is provided on the potential role of metabolites of the gut microbiome in stroke recovery through immunological repair processes, thus outlining an interesting scenario in which malnutrition and dysbiosis can intervene in the processes of plasticity and stroke outcome [106]. 

## 5. Impact of Malnutrition on Post-Stroke Neurorehabilitative Outcome

Malnutrition, as a pre-stroke existing condition or occurring during the post-acute ischemic injury, is significantly correlated with poor rehabilitative outcome [107,108,109]. The control of the trunk and its functional recovery are significantly impacted by the impaired functional motor recovery concerning actions such as standing and moving [9]. Hypomobility or bed rest and post-stroke fatigue syndrome, very common conditions in stroke patients, limit their compliance with rehabilitative treatments, namely, rehabilitative exercise, resulting in poor neurological recovery outcomes and motor functions [110,111,112,113]. 

Malnutrition and its associated sarcopaenia, as well as cardio-respiratory-muscle deconditioning, also worsen as a result. Notably, the vicious circle that arises between hypomobility and sarcopenia aggravates malnutrition, also occuring due to a quantitatively and qualitatively insufficient dietary intake, which, in turn, negatively affects the antioxidant and anti-inflammatory balance. 

Due to its pleiotropic property, nutritional intervention, which includes the improvement of the redox state, combined with rehabilitative aerobic physical activity, can contribute to diminishing stroke-associated inflammatory and oxidative statuses and to promoting better motor recovery in malnourished stroke patients [113,114]. Therefore, in order to prevent the burst of oxidative and inflammatory stress, which interfere with neural plasticity and neurological recovery processes, it is crucial to adopt adequate and pertinent nutritional behavior. Protein-energy malnutrition, whether originating before or after an acute stroke, is a risk factor for a worse outcome, prolonged hospital stay, higher frequency of respiratory and urinary infections, bedsores, and increased mortality rates at 3–6 months after stroke [115]. Malnutrition is also involved in post-stroke cognitive impairment, mainly in global cognition and frontal domain functions. In the subacute rehabilitation stroke setting, nutritional support based on increased supplementation of amino acids is associated with improved global cognitive outcomes due to the enhanced synthesis of neural proteins that favor axonal sprouting and new cortical connections. [114]. Malnutrition, particularly in elderly stroke patients, results in muscle mass loss, muscle fatty infiltration, and skeletal muscle atrophy in the affected limb(s). Such phenotypic skeletal muscle modifications, known as “stroke-related sarcopenia”, are recognized as useful predictors of limited mobility and poor rehabilitative outcomes [116]. On the contrary, neurorehabilitation treatment in elderly non-sedentary stroke patients with a proper nutritional status may promote skeletal muscle remodeling and predispose patients towards satisfying global motor and functional recovery [16]. Interestingly, a low BMI and low serum albumin levels lead to poor functional recovery. The combination of these parameters may be considered a valuable and prognostic marker of malnutrition, which may be more efficient than the single factor alone [117]. Obesity is the opposite of malnutrition. Evidence has shown that a high BMI may compromise functional recovery and that the recovery of functional independence in activities of daily living (ADL) in obese stroke patients after a neurorehabilitation intervention is related to decreased fat mass and catabolic processes [17]. According to the WHO recommendation concerning the BMI cut-off, a BMI value that falls within the range of a correct nutritional status may represent an independent prognostic factor for functional recovery after a stroke [118]. Although it has long been known that obesity is a real and effective risk for suffering a stroke, some studies have provided evidence on its positive influence on rehabilitative functional recovery, known as the so-called “obesity paradox” [119]. However, the assumption that “the fatter, the better” may be questionable. In fact, it remains to be considered that the evaluation of a patient’s BMI as a single index of stroke functional recovery may present some limits that are impossible to deduce from the evaluation of this single parameter, for instance, which component of body composition, fat mass, or fat free mas, is involved in the best rehabilitative functional recovery of obese stroke patients with respect to normal or underweight patients [17,120]. Findings from a sizable prospective cohort enrolled in the Feed Or Ordinary Diet (FOOD) study highlight that the nutritional status early after a stroke is independently associated with the long-term outcome [121]. Therefore, maintaining an adequate nutritional status or restoring it to levels matching the nutritional demands of post-stroke patients should be considered as significant requisites for achieving successful rehabilitation outcomes [1,2].

## 6. Nutritional Interventions in Post-Stroke Malnourished Patients Admitted to Rehabilitation

To prevent or stop the detrimental effects of poor nutritional status on the success of rehabilitation, malnutrition in stroke patients must be reversed or stopped. Nutritional strategies focused on planning dietary interventions for restoring impaired energy balance and protein synthesis, addressing mineral deficiencies, and lowering excess ROS and inflammatory mediators, link and complete the medical management of post-stroke patients, and could be essential to reducing some of the malnutrition-associated consequences that hinder rehabilitation programs [14,22]. Furthermore, the positive effects of a leucine-enriched amino acid supplement on muscle mass, muscle strength, and physical function were observed in post-stroke patients with sarcopenia performing low-intensity resistance training in addition to a post-stroke rehabilitation program [122]. Supplementation with a hyperproteic nutritional formula (250 kcal and 20 g of protein in addition to the baseline diet) in subacute stroke patients admitted to rehabilitative treatment seems to have enhanced neurological recovery, which was measured using the National Institute of Health Stroke Scale (NIHSS). Interestingly, the carbohydrate/protein intake ratio was directly related to the NIHSS score, while increased protein intake was associated with significant improvements evidenced by the inverse correlation between the protein intake and NIHSS score [123]. Although the question concerning resting the energy expenditure of stroke patients remains controversial, which is mainly due to the heterogeneity of the patients and methodologies used [124], there is general agreement that during rehabilitative treatment, an increased energy requirement due to an intensive regimen of mobility-associated activities is burdensome, especially in malnourished patients [125]. In a retrospective cohort study, it has been reported that the higher the energy levels at rehabilitation admission, the higher the Functional Independence Measure (FIM)’s efficiency and nutritional status improvement. It has also been observed that energy intake ≥ 26 kcal/kg/ day per ideal body weight is required to promote a greater increase in the improvement of ADL and nutritional status improvement [126]. A recent meta-analysis on the effects of nutritional supplementation on rehabilitation for stroke patients showed increased ADL and reduced incidence of infections, but no statistically significant effect on functional outcomes as well as disabilities, complications, and laboratory data [127]. Evidence was provided regarding the positive effects of antioxidant supplementation (vitamin C and E, polyphenols and flavonoids from fruits and vegetables, whole grains, and so on) on rehabilitation effectiveness owing to their neuroprotective properties [14,22,128]. It was demonstrated that the effectiveness of vitamin D supplementation on rehabilitation recovery is dependent on its vitamin-related roles as neuromuscular and neuroprotective factors and on its positive influence on bone mineral density [24,129]. Finally, evidence showed that omega-3 fatty acid supplementation was linked to improving rehabilitation outcomes [125]. 

## 7. Limitations

The results regarding the positive effects of antioxidant and anti-inflammatory food components or foods with respect to the rehabilitation outcome in malnourished stroke patients are not fully conclusive, because most studies only include a small number of participants and use multiple outcome measures (e.g., the Barthel Index, the modified Rankin Scale, the Functional Independence Measure, the National Institute of Health Stroke Scale, etc.) to assess patients’ recovery. Additionally, the heterogeneous assessment of malnutrition across centers and the non-uniform selection of stroke patients (based on age, gender, dysphagia or not, stroke severity, comorbidities, pharmacological treatment, etc.) might have prevented the generalization of the findings of each study. Notably, no data are available on the efficacy of supplementation after a patient’s reintegration into society and daily life. 

## 8. Five-Year Perspective

It has been shown through compelling evidence that translational research plays a crucial role in providing a more precise description of the recovery process that neurorehabilitation is able to achieve, even in elderly and weak patients. In order to promote recovery from neurological deficits, there is currently a great deal of interest in establishing the stage of the rehabilitation process at which dietary patterns can modulate and boost neuronal plasticity. The neural signals driving plasticity are modulated by gene expression, whose variability influences stroke patient’s capacity to respond to rehabilitation treatment and functional recovery. [67,130]. Nutrigenomics studies will be useful for understanding the mechanisms by which antioxidant and anti-inflammatory dietary patterns, with their neuroprotective properties, may activate cellular signaling pathways that modify neuronal plasticity. Nutritional systems biology is indeed an emerging approach to favor the evolution from traditional medicine toward personalized medicine (5P medicine: Predictive, Preventive, Participatory, Personalized, and Precision Medicines) [131]. Rehabilomics, a branch of the -omics sciences, uses the analysis and dosage of biomarkers as a tool to analyze the effectiveness of exercise and, afterwards, create a tailored therapeutic proposal for each patient. However, to date, biomarkers have found rather limited uses in the validation of rehabilitation treatments and in the evaluation of their therapeutic efficacy. Biomarkers can be used in the evaluation of the efficacy of a correct nutritional status for promoting the modulation of the oxidative balance during neuromotor recovery [132]. For that reason, in the future, rehabilomics should be considered as an interesting approach towards more comprehensively understanding the mechanisms and interactions between malnutrition, oxidative unbalance, neuroplasticity, and functional recovery [133]. 

Further research is required to understand how changes in the gut microbiota and the consequent chronic intestinal inflammation can determine the worsening of a patient’s nutritional status and the occurrence of a detrimental impact on neuroplasticity [134].

## 9. Conclusions

In the coming future, with a translational approach, it will be appropriate for neurorehabilitation clinical units to follow a structured and standardized assessment of the nutritional status and body composition of patients with neurological deficits to set specific dietary corrections. The assessment of nutritional status may be included as a routine tool in rehabilitation settings to verify the degree to which its modification can influence rehabilitative outcomes.

## Figures and Tables

**Figure 1 nutrients-15-00108-f001:**
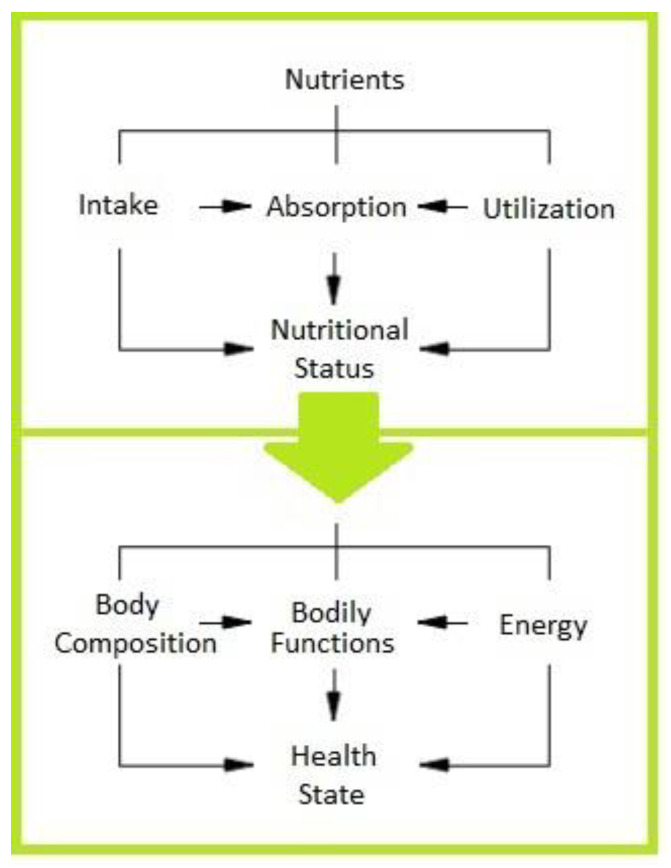
Nutritional status, measurable at a given time, is an integral part of the state of health. WHO defined the nutritional status “as the condition of the body, resulting from the balance of intake, absorption, and utilization of nutrients interacting with individual physiological and pathological status”. WHO, World Health Organization.

**Figure 2 nutrients-15-00108-f002:**
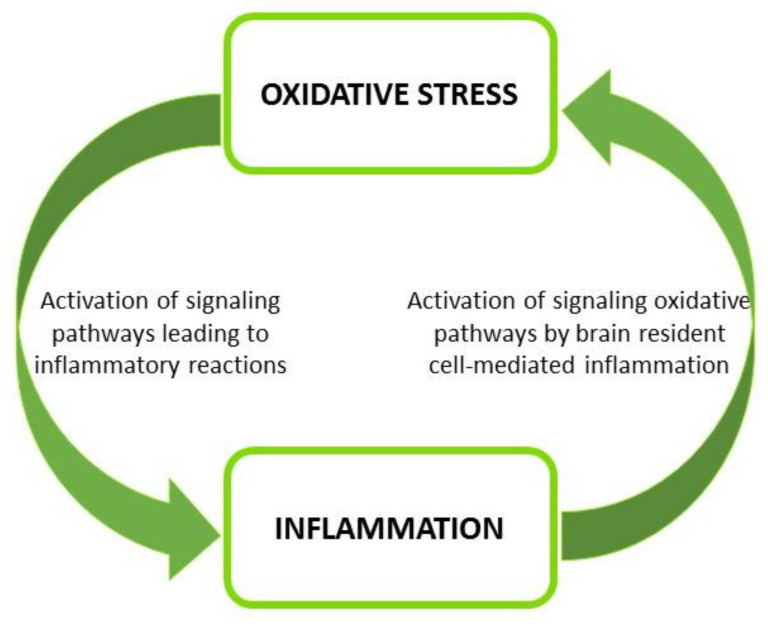
Cross-talk between oxidative stress and inflammation.

**Figure 3 nutrients-15-00108-f003:**
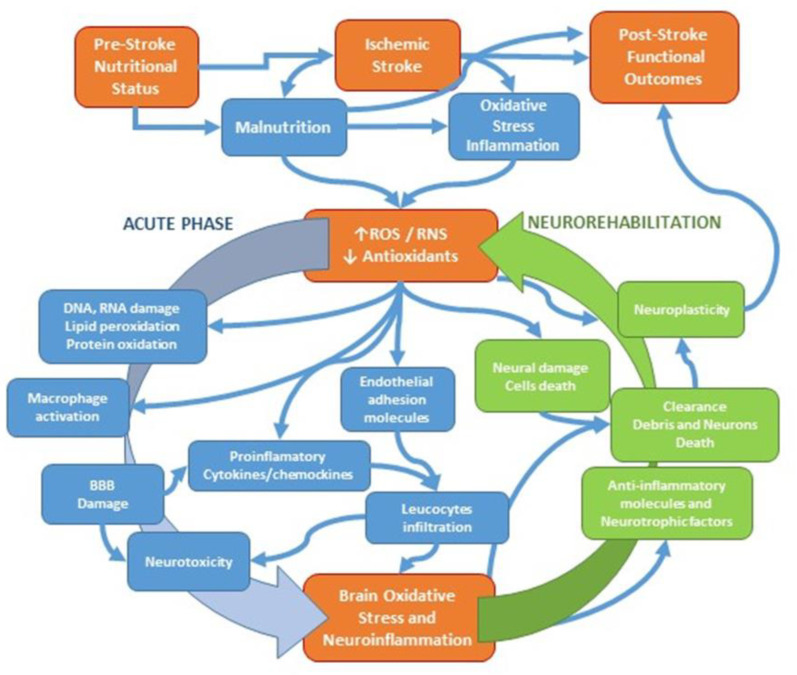
Schematic representation of stroke-associated oxidative stress and inflammation involvement in acute and rehabilitation phases. ROS, reactive oxygen species; RNS, reactive nitrogen species; BBB, blood–brain barrier; ↑, increase; ↓, decrease.

## Data Availability

Not applicable.

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
