# Peer review of "Influence of Oxidative Stress and Inflammation on Nutritional Status and Neural Plasticity: New Perspectives on Post-Stroke Neurorehabilitative Outcome"

_nutrients, 2022, doi:10.3390/nu15010108_

Round 1

Reviewer 1 Report

The author's made very long comprehensive review of nutrition, neuroinflammation and it's relationship with stroke. The topic is of interest and the article citation seems sufficient. However, generally each paragraph is too long and hard to follow. I recommend to divide each paragraph to have smaller number of sentences and made topic sentence for each paragraph.

Figure 3. Better to make clear which component makes negative/positive effect on each other. 

Author Response

Dear Editor in Chief,

Thank you for the opportunity to review this manuscript. We want to thank you and the Reviewers for your assistance with the manuscript entitled “INFLUENCE OF OXIDATIVE STRESS AND INFLAMMATION ON NUTRITIONAL STATUS AND NEURAL PLASTICITY: NEW PERSPECTIVES IN POST-STROKE NEUROREHABILITATIVE OUTCOME”. According to all the Reviewers and your suggestions, the manuscript has been revised, and all the requested corrections have been completed. Enclosed, you can find a point-by- point reply to the reviewers' comments. All changes to the manuscript have been highlighted in yellow. We hope that you and the Reviewers find our revision work satisfactory.

Dear Reviewer, 

The author's made very long comprehensive review of nutrition, neuroinflammation and it's relationship with stroke. The topic is of interest and the article citation seems sufficient. However, generally each paragraph is too long and hard to follow. I recommend to divide each paragraph to have smaller number of sentences and made topic sentence for each paragraph.

AU: We would like to thank the reviewer for the positive feedback regarding our review. According to the suggestion we have better divided the paragraphs with subheadings and removed some redundant sentences to increase readability.

Figure 3. Better to make clear which component makes negative/positive effect on each other. 

AU:. In regards to highlighting the positives and negatives effects: unfortunately there are mechanisms such as inflammation that are both positive (triggering neuroplasticity) and negative (if excessive), as clarified in the text. This makes a mere distinction between what is positive and what is negative difficult. Thanks for the comment, we have increased the readability of the figure by making the two acute and rehabilitation phases clearer.

Reviewer 2 Report

This review paper titled "Influence of Oxidative Stress and Inflammation on Nutritional Status and Neural Plasticity: New Perspectives in Post-Stroke Neurorehabilitative Outcome", aims to present the issue of malnutrition as a pre-stroke condition as a serious complication during rehabilitative treatment as it leads to poor rehabilitation outcomes. It is an interesting paper. The paper is well written. The authors are able to give a five-year perspective, addressing limitations and possible solutions. How can the diet be changed? 

Author Response

Dear Editor in Chief,

Thank you for the opportunity to review this manuscript. We want to thank you and the Reviewers for your assistance with the manuscript entitled “INFLUENCE OF OXIDATIVE STRESS AND INFLAMMATION ON NUTRITIONAL STATUS AND NEURAL PLASTICITY: NEW PERSPECTIVES IN POST-STROKE NEUROREHABILITATIVE OUTCOME”. According to all the Reviewers and your suggestions, the manuscript has been revised, and all the requested corrections have been completed. Enclosed, you can find a point-by- point reply to the reviewers' comments. All changes to the manuscript have been highlighted in yellow. We hope that you and the Reviewers find our revision work satisfactory.

---Reviewer

This review paper titled "Influence of Oxidative Stress and Inflammation on Nutritional Status and Neural Plasticity: New Perspectives in Post-Stroke Neurorehabilitative Outcome", aims to present the issue of malnutrition as a pre-stroke condition as a serious complication during rehabilitative treatment as it leads to poor rehabilitation outcomes. It is an interesting paper. The paper is well written.

AU: We would like to thank the reviewer for the positive judgement regarding our review.

The authors are able to give a five-year perspective, addressing limitations and possible solutions. How can the diet be changed?

AU: Thanks for the helpful comments. In accordance with the commentary, we have added subparagraphs regarding the limitations and the 5-year perspective. 

Round 2

Reviewer 1 Report

The left edge of Fig. 3 was lacking.

Author Response

Dear Reviewer, 

We want to thank you again for the positive opinion and for the time spent. According to your comment we have replaced Fig.3. We have also reduced the number of references according to the editor's comment. We are doing a professional English review.